# FPGA Implementation of Complex-Valued Neural Network for Polar-Represented Image Classification

**DOI:** 10.3390/s24030897

**Published:** 2024-01-30

**Authors:** Maruf Ahmad, Lei Zhang, Muhammad E. H. Chowdhury

**Affiliations:** 1Faculty of Engineering and Applied Science, University of Regina, Regina, SK S4S 0A2, Canada; mah370@uregina.ca; 2Department of Electrical Engineering, Qatar University, Doha 2713, Qatar; mchowdhury@qu.edu.qa

**Keywords:** image classification, complex-valued neural network, FPGA implementation, CVNN on FPGA

## Abstract

This proposed research explores a novel approach to image classification by deploying a complex-valued neural network (CVNN) on a Field-Programmable Gate Array (FPGA), specifically for classifying 2D images transformed into polar form. The aim of this research is to address the limitations of existing neural network models in terms of energy and resource efficiency, by exploring the potential of FPGA-based hardware acceleration in conjunction with advanced neural network architectures like CVNNs. The methodological innovation of this research lies in the Cartesian to polar transformation of 2D images, effectively reducing the input data volume required for neural network processing. Subsequent efforts focused on constructing a CVNN model optimized for FPGA implementation, emphasizing the enhancement of computational efficiency and overall performance. The experimental findings provide empirical evidence supporting the efficacy of the image classification system developed in this study. One of the developed models, CVNN_128, achieves an accuracy of 88.3% with an inference time of just 1.6 ms and a power consumption of 4.66 mW for the classification of the MNIST test dataset, which consists of 10,000 frames. While there is a slight concession in accuracy compared to recent FPGA implementations that achieve 94.43%, our model significantly excels in classification speed and power efficiency—surpassing existing models by more than a factor of 100. In conclusion, this paper demonstrates the substantial advantages of the FPGA implementation of CVNNs for image classification tasks, particularly in scenarios where speed, resource, and power consumption are critical.

## 1. Introduction

Recently, Generative Artificial Intelligence (GAI) [1] technologies have surged to the forefront, with tools like ChatGPT [2] and AI-powered image and video generators [3] like MidJourney [4] dominating the conversation. The core of these visual generators lies in image processing and classification, serving as the backbone of this AI-driven revolution. These breakthroughs have been made possible by the remarkable progress in artificial neural networks applied to image and video processing [5]. However, this progress has come at the cost of increased computational complexity. The amount of layers and neurons in each layer required for state-of-the-art deep models has grown significantly, often involving millions of parameters and billions of operations to achieve human-level accuracy.

Simultaneously, the growth of the Internet of Things (IoT) and embedded systems has led to an escalating demand for neural network models to perform various tasks. However, the computational demands of deep neural networks present challenges when deploying them on low-power embedded platforms with limited computational and power resources [6,7].

To address these challenges and enhance the efficiency of neural network algorithms, particularly in terms of reducing computational costs, energy consumption, and resource usage, multiple strategies have emerged. One approach focuses on reducing the theoretical number of basic operations required in neural network computations through algorithmic innovations. Simultaneously, another direction aims to improve neural network algorithms using hardware accelerators, such as Application-Specific Integrated Circuit (ASIC) and FPGA designs [6,7].

In our work, we explore both of these approaches. We present innovative image preprocessing methods tailored for neural network models and introduce a hardware accelerator model designed to reduce computational costs in neural networks and optimize energy and resource utilization in hardware systems. This research endeavors to contribute to the ongoing efforts aimed at making neural network applications more efficient and sustainable, addressing the challenges posed by increasing computational demands.

The efficacy of image classification models hinges not only on the sophistication of neural network architectures but also on the quality of the input data and the performance of the processing pipeline. Traditionally, image classification processes have operated in Cartesian coordinates (x, y), where 2D images are serialized for neural network input. However, this conventional approach poses challenges. Applying frequency analysis methods like the Fourier transform to serialize two-dimensional (2D) images can inadvertently lead to the omission of significant spatial data. This pertains to the positional information of the pixels and the contextual interactions between neighboring pixels, which can be diminished in the serialization transition.

Recent research [8] has addressed this limitation by introducing a novel preprocessing pipeline that transforms standard image datasets into a polar coordinate representation. This transformation is inspired by the recognition that polar coordinates, defined by radial distance (r) and angular displacement (θ) from a reference point, provide a more natural representation for circular and radial patterns. This transformation effectively retains the spatial information inherent in the pixel arrangement of the original image. By encoding images in polar coordinates, it aims to exploit these advantages and enhance the classification accuracy of image datasets. However, the research has not yet ventured into the practical application of this method in real-world image classification, despite its initial focus on constructing an Spiking Neural Network (SNN) model. To overcome the existing gap in research, our study focuses on replicating the process of converting 2D images from Cartesian to polar coordinates. Subsequently, we apply this transformation technique to the challenge of classifying the Modified National Institute of Standards and Technology (MNIST) [9] handwritten digit dataset using artificial neural networks.

An essential aspect of the preprocessing pipeline involves the application of the Discrete Fourier Transform (DFT) to the polar-transformed images. The DFT is a powerful mathematical tool for decomposing signals into their constituent frequency components, and it has found extensive use in image analysis. In our approach, we leverage the DFT to extract both magnitude and phase information from the polar-represented images. The utilization of complex exponentials within the DFT allows us to capture nuanced variations in pixel values and relationships, thus preserving essential spatial data.

What sets our research apart is the integration of Complex-Valued Neural Networks (CVNNs) into the image classification framework. Unlike traditional Real-Valued Neural Networks (RVNNs), CVNNs are tailored to handle complex-valued data, such as the output of the DFT. By treating the DFT outputs as complex numbers, we effectively harness the rich information embedded in both the real and imaginary parts. This nuanced approach promises to provide a more holistic understanding of the input data, potentially leading to improved classification accuracy.

While the theoretical advantages of CVNNs in image classification have been explored in the literature, there is a notable gap in the practical implementation of such networks, particularly on resource-constrained platforms. Therefore, our research extends beyond theoretical exploration to encompass practical deployment. We aim to implement the trained CVNN model on Field-Programmable Gate Array (FPGA), capitalizing on the parallel processing capabilities intrinsic to FPGA architecture. The FPGA implementation offers the potential for real-time classification with significantly reduced computational resources and power requirements compared to traditional CPU-based sequential computing and GPU-based parallel computing.

### Research Objectives

The primary goal of this research is to explore the efficacy of polar coordinate representation of 2D image data and its impact on complex-valued neural networks (CVNNs) and real-valued neural networks (RVNNs) in image classification tasks. A key focus is the comparative analysis of these neural networks in handling complex-valued inputs and the subsequent implementation on FPGA to assess resource utilization, power efficiency, and inference speed. The goal is to demonstrate the practical advantages of FPGA acceleration for real-time image classification, thus achieving faster classification while consuming fewer resources and power.

## 2. Related Work

Within the domain of image classification, conventional neural network models like feedforward neural networks [10], CVNNs, Recurrent Neural Networks (RNNs) [11], and deep neural networks [12] have primarily been designed to handle real-valued data. However, the growing prevalence of complex-valued data sources [13,14,15] such as complex-valued MRI images, SAR (Synthetic Aperture Radar) images, sonar images, Optical Coherence Tomography (OCT) images, as well as sound and wave signals, has spurred the need for specialized neural network models capable of directly processing complex-valued inputs. Consequently, researchers have responded by developing a range of models tailored to meet this specific demand.

As the capabilities of neural network models continue to grow in complexity, and the data they process become increasingly vast, the computational demands and time required for tasks have surged [6]. Consequently, there is a pressing need to discover solutions that can enhance the speed and throughput of neural networks while minimizing energy consumption. This has led to the emergence of hardware accelerators as a pivotal area of research focus [7]. Accelerators utilizing GPUs, FPGAs, and ASICs have garnered attention for their potential to meet the performance requirements of deep learning tasks. While GPU-based models have shown considerable performance, their applicability in power-sensitive embedded devices remains a challenge, primarily due to their higher energy consumption. In contrast, FPGAs have gained prominence for their remarkable energy efficiency [16,17], flexibility, and shorter development periods compared to ASICs. With robust parallel computing capabilities and reduced energy consumption, FPGAs have risen to prominence in the field of hardware acceleration for deep learning. These reconfigurable devices enable engineers to simulate digital circuits efficiently, paving the way for enhanced neural network computation. Unlike CPUs, which face inherent structural limitations when processing vast amounts of data, FPGAs offer a versatile solution with virtually limitless reconfigurable logic, enabling the creation of tailored accelerators for a multitude of applications. This inherent adaptability, combined with their capacity for parallel processing and pipeline optimization, positions FPGAs as a compelling choice for fast and energy-efficient neural network model implementations [7].

In recent research from our lab, Zahng et al. [18] presents an energy-efficient spiking neural network (SNN) designed and implemented on FPGA, emphasizing lower power consumption and minimal accuracy loss. The approach utilizes rate coding to map Artifical Neural Network (ANN) parameters to SNNs efficiently, yielding a power efficiency of 8841.7 frames/watt with minimal accuracy degradation. The system sets a new performance standard, achieving an impressive 90.39% accuracy rate, outperforming conventional SNN benchmarks.

Several studies [19,20,21] have investigated hardware accelerators for MNIST classification using neural networks, primarily centered on CNNs. These studies also provide comparisons of speed and resource utilization in contrast to CPUs or GPUs.

As of May 2023, we have identified just one instance of a CVNN model implemented on FPGA [22]. The study introduces ComplexNet, a deep convolutional CVNN for channel estimation (CE) in 5G Orthogonal Frequency Division Multiplexing (OFDM) communication systems. It demonstrates that ComplexNet enhances CE accuracy and offers a lightweight FPGA implementation, significantly reducing power consumption compared to CPU and GPU platforms.

Notably, to the best of our knowledge, no prior implementations of CVNNs on FPGA for MNIST dataset classification have been found in our research.

The remaining sections of the paper are organized as follows:

Section 3: In this section, we provide an overview of the Cartesian to polar coordinate representation for 2d images.

Section 4 and Section 5: In these sections, we provide an account of how the CVNN is implemented for preprocessed image data. We also compare its performance with its real-valued neural network counterparts.

Section 6 and Section 7: These sections offer an explanation of how the hardware accelerator is implemented for CVNN using FPGA. We present an analysis of its performance compared to running the model on environmental setups like CPU and GPU. We also compare it against existing research.

Section 8: The final section serves as a summary of this research. It outlines discussions, highlights any limitations encountered during research, suggests areas for future work, and provides concluding remarks.

## 3. Cartesian to Polar Coordinate Representation

In the present study, the MNIST database of handwritten digit images was employed. Recognized as a benchmark dataset for neural network modeling and computer vision, the MNIST dataset consists of handwritten digits from 0 to 9. This dataset is commonly utilized for the training and assessment of classification algorithms. As depicted in Figure 1A, the dataset offers representative samples of each numeral from 0 through 9. It has 60,000 training and 10,000 testing samples. It is noteworthy that the pixel values within these images span from 0 to 255, indicating the grayscale intensity.

In our study, we explored a methodology from a published paper [8] that transitions MNIST images from the traditional Cartesian coordinate system (x and y) to the polar coordinate system, defined by magnitude (ρ) and angle (θ), using complex number representation. This technique effectively captures the spatial characteristics of pixel locations and their relationships during serialization. Figure 1 shows a sample of 0–9 original MNIST digits, polar coordinate representation, and a serialized view after discrete Fourier transformation.

To convert from Cartesian to polar coordinates, a series of steps is undertaken, including a logarithmic transformation, contour highlighting, downsampling, and the actual Cartesian to polar coordinate transformation. The resulting polar-represented MNIST digits can be seen in Figure 1B. Following this, zero padding and decimation techniques are employed to select a specific number of data points. Figure 1C displays the polar-represented images with 128 data points, while Figure 1D illustrates the images with 64 data points. Finally, the polar-represented images undergo serialization through discrete Fourier transform (DFT), which captures both the amplitude and phase of specific frequency components within the original sequence, as depicted in Figure 1E,F. The detailed process for these steps is discussed in Appendix A.

## 4. Complex-Valued Neural Network (CVNN) Implementation

 CVNNs represent a specialized class of neural networks that operate on complex numbers, encompassing both real and imaginary components [23]. While conventional neural networks primarily deal with real-valued data, CVNNs have gained prominence in applications where data inherently exhibit both magnitude and phase information, especially those involving signals, waves, or data with phase information.

In the context of this study, the relevance of CVNNs stems from their intrinsic capability to effectively process complex-valued data. The research is focused on the preprocessed MNIST dataset, which undergoes a transformation from Cartesian to polar coordinates, followed by serialization using the discrete Fourier transform (DFT). This transformation equips the dataset with complex numbers, enabling a more compact representation while retaining critical phase information, which is essential for character recognition tasks [24].

### 4.1. Training Procedure and Hyperparameters

We developed a CVNN model using the cvnn library, which is open source and accessible on GitHub (https://github.com/NEGU93/cvnn accessed on 28 January 2024). Detailed documentation can be found on Read the Docs (https://complex-valued-neural-networks.readthedocs.io/en/latest/index.html accessed on 28 January 2024). This library is distributed under the MIT License, promoting flexibility and ease of use for the community.

Our CVNN architecture is designed as a feedforward model, constructed using TensorFlow’s renowned Sequential API [25]. The network structure consists of an initial input layer, followed by two dense layers, carefully crafted to efficiently process and manage complex-valued data. The programming language and libraries used for building the model are as follows in Table 1.

#### 4.1.1. The Parameters for the Feedforward Model Are Configured as Follows

The model begins with a “ComplexInput” layer with an input shape of 128 or 64.Subsequently, a “ComplexDense” layer is added with a varying number of neurons, depending on the specific model. The “crelu” (Equation (Equation 17), Appendix B) activation function is chosen, and the layer is initialized with the ’ComplexGlorotUniform’ initializer.The final layer in our model is another “ComplexDense” layer with 10 neurons for the classification of ten different MNIST handwritten digits. It utilizes the “cart_softmax” activation function and is initialized with the “ComplexGlorotUniform” initializer.

#### 4.1.2. The Training Parameters for the CVNN Are Configured as Follows

Optimizer: We used the “Adam” optimizer, a well-known optimization algorithm that adapts the learning rate during training.Loss Function: Our model employs the “ComplexAverageCrossEntropy” loss function, which applies Categorical Cross-entropy to both the real and imaginary parts separately and then averages the results.Metrics: Model performance is evaluated using the “ComplexCategoricalAccuracy” metric, which measures how often predictions match one-hot labels.Training: The model is trained using the “fit” method with a batch size of 32 and a specified number of epochs (in our case, 50). Both training data (“train_images” and “train_labels”) and validation data (“val_images” and “val_labels”) are provided for this method.Learning Rate: Our implementation does not specify any learning rate schedules, so the learning rate defaults to the value set by the “Adam” optimizer, which is “0.001”.

### 4.2. Experimental Setup

In this section, we outline the comprehensive experimental setup for our study, conducted on Google Colab, utilizing the Ubuntu 22.04.2 LTS environment with abundant system resources. We detail the hardware and software configurations, as well as the dataset and model variations considered.

#### 4.2.1. Hardware and Software Environment

Our experiments were conducted on Google Colab, specifically version 1.0.0, which offers a robust cloud-based environment for machine learning tasks. The underlying specifications of the environment are as follows in Table 2.

#### 4.2.2. Dataset and Model Variations

Our research investigates the impact of varying the number of data points for input within the context of polar-represented serialized MNIST digit datasets. In order to conduct a comprehensive analysis of the performance of our models, we developed two independent models with varying input configurations shown in Table 3.

The deliberate manipulation of the quantity of data points enables us to thoroughly assess the effectiveness of our models across various input configurations.

#### 4.2.3. Dataset Details

The dataset utilized in this study consists of serialized MNIST digit pictures rendered in polar coordinates. This representation offers the benefit of reducing the number of data points required for input. The dataset encompasses a range of key statistical measures, which are as follows in Table 4.

The dataset at our disposal is highly suitable for evaluating the efficacy of our models, hence facilitating the derivation of significant comparisons and insights.

In brief, the experimental configuration for our study is established within a reliable Google Colab environment that offers substantial computational capabilities. In this study, we examine the influence of various input configurations on the performance of a model. Our analysis is based on a dataset consisting of serialized MNIST digit pictures stored in polar form. The purpose of this configuration is to enable thorough examinations and offer significant observations regarding the performance of our models.

## 5. Results and Performance Evaluation

This section provides an overview of the experimental results and performance evaluation of the CVNNs when trained on the preprocessed MNIST dataset. Our team offers complete insights into our model’s behaviour through the provision of thorough visualizations, encompassing accuracy trends, training curves, and loss plots.

In the present study, we utilize two separate models of CVNNs. The initial model, referred to as CVNN_Polar_128, is specifically designed to handle a dataset that is serialized in the polar representation and consists of 128 data points. On the other hand, the second model is designed to accommodate a comparable dataset, albeit with a diminished count of 64 data points, and is appropriately denoted as CVNN_Polar_64.

### 5.1. Accuracy Metrics

To assess the model’s performance comprehensively, we analyze the accuracy trends for both of the models with the serialized polar-represented testing datasets.

#### 5.1.1. Graph Interpretation

The graph in Figure 2 visualizes the testing accuracy of two CVNN models, CVNN_Polar_128 and CVNN_Polar_64, against the number of neurons in their hidden layers. The horizontal axis delineates the neuron count, ranging from 5 to 100, while the vertical axis marks the achieved testing accuracy, expressed as a percentage. A vertical dotted line at “20 Neurons” highlights the chosen neuron count, offering a visual cue for the selection. Both models demonstrate a general upward trend, suggesting that increasing the neuron count positively impacts the accuracy, up to a certain threshold.

#### 5.1.2. Justification for Selection of 20 Neurons

From an optimization standpoint, selecting 20 neurons for the hidden layer of both models appears judicious for several reasons:**Balanced Complexity and Performance**: At 20 neurons, both models achieve a substantial increase in accuracy compared to lower neuron counts, without the added computational overhead of higher counts. This makes the models efficient without compromising on performance.**Diminishing Returns**: While further increments in neuron count do lead to accuracy improvements, the gains become marginal. For instance, the leap from 20 to 100 neurons results in an increase of just over 2% for both models, which may not justify the associated computational cost and potential overfitting risks.**Generalization**: A model with fewer neurons is less prone to overfitting. With 20 neurons, **CVNN_Polar_128** achieves an accuracy of 88.3%, and **CVNN_Polar_64** attains 87%. These figures highlight efficient model architectures given the neuron count.**Computational Efficiency**: Neural networks with fewer neurons train faster and require less memory. From a practical standpoint, especially in real-time applications or scenarios with limited computational resources, a leaner model is advantageous.

In the conducted experiments, two CVNN models were trained on the processioned MNIST dataset: CVNN_Polar_128 with 128 data points and CVNN_Polar_64 with 64 data points. Details of other performance evaluations, visualized through accuracy trends, training curves, and loss plots, are provided in Appendix C.

### 5.2. Comparison with Real-Valued Neural Networks

In this section, we compare the performance of CVNNs with RVNNs in handling complex data, specifically, the polar-transformed MNIST digit dataset. We delve into accuracy metrics between these two types of networks, shedding light on the advantages of employing CVNNs for complex data.

#### 5.2.1. Models’ Overview

**RVNN_Raw_MNIST**: A real-valued neural network that operates directly on the raw MNIST dataset without any preprocessing. This model serves as a benchmark, providing a standard to which other models can be compared.**RVNN_Polar_128 and RVNN_Polar_64**: Real-valued adaptations designed to handle the polar-transformed dataset. To accommodate the complex nature of the data, these models separate and concatenate the real and imaginary parts, effectively doubling their input neuron requirements.

#### 5.2.2. Processing Complex Data in Real-Valued Networks

By segregating the real and imaginary components of the serialized polar-transformed MNIST dataset and then concatenating them, the models were furnished with doubled input fields. This ensured that the entirety of the complex data was captured, albeit in a format palatable to real-valued networks. For instance, the RVNN_Polar_128 model, designed for 128 complex data points, required 256 neurons in its input layer to accommodate both the real and imaginary parts.

When comparing the RVNN and CVNN models, both were configured with an identical number of hidden layer neurons, set at 20 for this analysis.

Additionally, other parameters, including the number of output layer neurons, batch size, and epochs, were kept uniform across both models for a consistent evaluation.

#### 5.2.3. Performance Insights

**Benchmark Performance**: While the RVNN model operating on the original MNIST dataset set a high standard with an accuracy of 96%, our focus was primarily on the performance gains achieved through polar transformation.**Complex-Valued vs. Real-Valued on Polar Data**: As hypothesized, the CVNN_Polar_128 model, attaining an accuracy of 88.3%, outperformed its RVNN counterpart, RVNN_Polar_128, which secured 87.5%, as shown in Figure 3. This 0.8% differential underscores the inherent advantage of CVNNs when processing polar-transformed data. The separation of real and imaginary components in RVNNs, while necessary, may lead to the omission of valuable interplay between these components, a nuance that CVNNs naturally capture.**Data Efficiency through Polar Transformation**: The polar-transformed models, even with reduced data points, achieved commendable accuracies. The slight performance trade-offs were balanced by the benefits of reduced computational requirements and energy consumption.

#### 5.2.4. Comparison with Contemporary Research

Jose Agustin Barrachina, in his implementation of the CVNN model [26], conducted MNIST handwritten digit classification. He transformed the original MNIST dataset from its real-valued version to a complex-valued version using TensorFlow’s tf.cast function and tf.complex64 data type. This conversion resulted in each pixel of the image comprising both real and imaginary components. Following testing, he achieved an impressive 99% accuracy for the MNIST dataset. In our comparison table, we refer to Jose Agustin Barrachina’s model as CVNN_JAB.

While his model achieved higher accuracy, it came at the cost of increased computational complexity. This was due to the inclusion of all 784 data points for each MNIST image. In contrast, our model only utilized 128 data points, resulting in reduced computational demands. This makes it a more efficient choice for hardware accelerator implementation in resource- and energy-constrained environments.

According to the PapersWithCode.com website, as of today, the highest accuracy achieved in MNIST classification is 99.83% by Byerly, A. et al. [27].

## 6. FPGA Implementation of CVNNs

In this section, we transition from the theoretical aspects discussed in the previous sections, which focused on the polar representation of 2D images, serialization techniques, and their application to neural networks (CVNN and RVNN) for MNIST handwritten digit classification. Here, our focus shifts to the hardware implementation of these neural network’s inference models, with a specific emphasis on CVNNs, using FPGA.

### 6.1. Complex-Valued Neural Network Inference Model

In the previous sections, we delved into the intricacies of forward and backward propagation within the CVNNs. As we transition into this section, our primary focus is on offering a succinct recapitulation of forward propagation. This is imperative for a holistic understanding of how we implemented the CVNN inference model on FPGA.

Our implementation of the CVNN inference model leverages the weights and biases from a pretrained model. For a visual representation of our CVNN model’s architecture, one can refer to Figure 4. It is pivotal to note that all the parameters, including input *X*, weight *W*, bias *b*, and output *Y*, are complex-valued in this architecture.

Complex numbers in our context are typically depicted as a+bi, where *a* signifies the real part, *b* stands for the imaginary part, and *i* is the imaginary unit.

To elucidate further, let us dissect the model’s operations:**Weighted Sum in Hidden Layer**: The first step in our forward propagation is the computation of the weighted sum for each neuron in the hidden layer. This is achieved by linearly combining the complex-valued inputs with their respective weights, adding the complex-valued biases subsequently. For instance, the weighted sum $Z_{1}$ for the first neuron in the hidden layer can be represented as:
(1)$$Z_{1}=X_{1}\times Wh_{1}+X_{2}\times Wh_{2}+\ldots +X_{n}\times Wh_{n}+bh_{1}\;$$
where *X* refers to complex inputs, Wh refers to the complex weights in the hidden layer, and bh denotes the complex biases in the hidden layer.**Activation in Hidden Layer**: Following the computation of the weighted sum, we introduce non-linearity through the Complex ReLU (CReLU) activation function. This function, applied to each neuron’s weighted sum, separates the real and imaginary components. It then rectifies negative values from both parts. For the first neuron, the activation is:
(2)$$H_{1}=\mathrm{CReLU}\left(Z_{1}\right)=\max\left(0,\mathrm{Real}\left(Z_{1}\right)\right)+i\times \max\left(0,\mathrm{Imag}\left(Z_{1}\right)\right)$$
where *H* is the output of a hidden layer neuron after activation.**Weighted Sum in Output Layer**: The outputs from the hidden layer are then used to compute the weighted sum for each neuron in the output layer. This involves multiplying each output from the hidden layer by the respective weights of the output neurons and adding the corresponding biases.
(3)$$O_{1}=H_{1}\times Wo_{1}+H_{2}\times Wo_{2}+\ldots +H_{n}\times Wo_{n}+bo_{1}$$
where Wo are the weights in the output layer, bo are the biases in the output layer, and *O* is the weighted sum of a neuron in the output layer.**Activation in Output Layer**: Finally, the CReLU activation function is once again applied to the weighted sums from the output layer to yield the final complex-valued outputs of the model. Using the first output neuron as an example:
(4)$$Y_{1}=\mathrm{CReLU}\left(O_{1}\right)=\max\left(0,\mathrm{Real}\left(O_{1}\right)\right)+i\times \max\left(0,\mathrm{Imag}\left(O_{1}\right)\right)$$

In the original CVNN model, the softmax activation function was employed for the output layer, providing a probabilistic interpretation of the model’s predictions. However, when transitioning to FPGA implementation, it is imperative to strike a balance between computational accuracy and hardware efficiency. Given this consideration, we opted for the CReLU activation function in our project. This choice not only streamlines the FPGA implementation but also ensures a robust performance while simplifying the overall computational complexity.

In our pursuit of drawing a comprehensive comparison with the CVNN model, we implemented an RVNN inference model on FPGA. The underpinnings of this implementation draw many parallels to the CVNN model.

For the RVNN, we took a strategic approach by separating the complex-valued inputs into their real and imaginary components. These separated components were then treated as independent real-valued inputs. Consistent with the RVNN paradigm, the weights, biases, and activation functions were all real-valued. Specifically, the ReLU activation function was employed for the RVNN, described by the equation:(5)$$H_{n}=\mathrm{ReLU}\left(Z_{n}\right)=\max\left(0,Z_{n}\right)$$

This function effectively nullifies negative values, allowing only positive activations to propagate through the network. By juxtaposing the CVNN and RVNN models, we aim to provide a holistic understanding of their respective performances and intricacies on FPGA platforms.

### 6.2. FPGA Implementation of Inference Model

In our previous discussions, we thoroughly examined the mathematical foundations inherent to the inference models of neural networks that were central to our research. Transitioning from theory to application, the linchpin of the FPGA implementation of these models is the precise crafting of VHSIC Hardware Description Language (VHDL) modules that correspond to each step and mathematical equation. At the heart of the architectures of both the CVNNs and RVNNs lie several core modules: adders, multipliers, and the specific activation functions: the complex rectified linear unit (CReLU) and the traditional rectified linear unit (ReLU).

For this FPGA-centric endeavor, we predominantly utilized Vivado v2021.1 64-bit for design and synthesis, combined with VHDL for hardware description and programming.

The accompanying Figure 5B offers a schematic representation of a singular neuron’s FPGA realization.

#### 6.2.1. Adder

**Real-valued adder:** The adder module for real numbers in a digital FPGA environment is implemented using fixed-point arithmetic. In fixed-point representation, every number is represented as an integer and a fractional part. Mathematically, given two fixed-point numbers *A* and *B*, the summation *S* is given by:(6)S=A+B

**Complex-valued adder:** Complex numbers consist of real and imaginary components. Thus, for two complex numbers $C_{1}=a+bi$ and $C_{2}=x+yi$, the resultant *R* after addition is:(7)R=(a+x)+(b+y)i

#### 6.2.2. Multiplier

**Real-valued multiplier:** In the realm of fixed-point arithmetic, when two numbers *P* and *Q* are multiplied, the result *M* is:(8)M=P×Q

**Complex-valued multiplier:** The multiplication of two complex numbers $C_{1}=a+bi$ and $C_{2}=x+yi$ results in:(9)R=(a×x−b×y)+(a×y+b×x)i

#### 6.2.3. Activation Functions

Activation functions introduce non-linearity into neural networks, allowing them to capture intricate patterns and make complex decisions.

**Rectified linear unit (ReLU):** One of the most widely adopted activation functions in our research, ReLU is mathematically defined as:(10)f(x)=max(0,x)
where *x* is the input to the neuron.

For FPGA implementation using fixed-point arithmetic, we employed a simple comparison of the input with zero, choosing to either return the input or zero based on the outcome of this comparison.

**Complex rectified linear unit (CReLU):** In our exploration of CVNNs, we utilized CReLU, which operates on complex numbers by applying the ReLU function independently to both the real and imaginary parts of the input. Given a complex number C=a+bi, the output after CReLU, *R*, is:(11)R=max(0,a)+max(0,b)i

Our VHDL implementation is analogous to the one for ReLU but distinctly applies the function to both real and imaginary components.

Complex-valued multiplier and complex rectified linear unit activation function schematic diagrams implemented on Vivado are shown in Figure 6.

### 6.3. Fixed-Point Quantization for Neural Network Inference on FPGA

Fixed-point representation is a widely used approach for representing real numbers in digital systems, especially in FPGA implementations. Unlike floating-point representation, which dynamically adjusts precision and range, fixed-point representation assigns a set number of bits to both the integer and fractional parts of a number. This methodology presents a harmonious balance between precision, range, and computational demand, rendering it particularly suitable for high-speed and resource-limited FPGA designs.

In the realm of neural network inference, the choice of fixed-point representation becomes pivotal. It determines not only the network’s accuracy performance but also the FPGA implementation’s efficiency. Both range (the span of representable numbers) and precision (the smallest distinguishable difference between numbers) emerge as vital considerations.

For our specific neural network inference model, the dynamic range of the data lies between −120 and 120. It is imperative that our chosen numerical representation can accommodate this range. Additionally, to preserve model accuracy, the system must achieve a precision capable of differentiating values with a minimum difference of 0.01.

Given these prerequisites, a 16-bit fixed-point representation was our chosen configuration. This selection permitted an even bit distribution, allotting 8 bits to the integer segment and 8 bits to the fractional segment. The rationale behind this is twofold:**Integer Part**: Employing an 8-bit integer representation (with one bit reserved for sign) enables the system to represent values spanning from −128 to 127. This adequately covers our anticipated data range from −120 to 120, ensuring overflow is a non-issue.**Fractional Part**: An 8-bit fractional part translates to a resolution of $2^{-8}$, approximately equal to 0.0039. This precision surpasses our stipulated minimum of 0.01, guaranteeing that our system can depict values with the necessary precision and, in turn, safeguarding our model’s inferential accuracy.

This fixed-point configuration aligns seamlessly with the distinct requirements of our neural networks (CVNN and RVNN alike). Moreover, it taps into the inherent strengths of FPGAs, such as computational parallelism and efficient arithmetic operations. Through this astute choice of representation, we ensure the fidelity of our implemented neural network models while reaping the benefits of the speed and resource efficiencies native to FPGA-based designs.

### 6.4. FPGA Structure for the Proposed Neural Network Systems

In our research, we generated two datasets from the serialized polar representations of MNIST images. The first set, polar-transformed MNIST with 64 input fields, is referred to as PT_MNIST_64. The second, with 128 input fields, is termed PT_MNIST_128. Considering the limited number of I/O pins on our target FPGA development board, we optimized the system to process these input fields in batches, handling eight fields for each batch per clock cycle, to address this limitation. Consequently, for the PT_MNIST_64 dataset, which contains 64 input fields per sample, the system requires 8 clock cycles, and for the PT_MNIST_128 dataset with 128 input fields per sample, it necessitates 16 clock cycles to fully process all input fields.

Each dataset comprises complex-valued data, entailing both real and imaginary components. Consequently, the required I/O pin count doubles. Adopting a 16-bit representation, the input field pin requirement is calculated as 8×16×2=256.

The flow of data within the system is depicted in Figure 5A. In the initial clock cycle, the system multiplies the first eight input values with their corresponding hidden layer weights. This weighted sum is subsequently directed to a buffer adding the weighted sum of the current batch (CB) to the current register (Reg). Registers are constructed using FF and LUTs. Notably, no dedicated memory blocks are used for data storage during processing in these registers. This register buffer retains the summation, waiting until it assembles the complete set of input fields from a given sample. For instance, in the case of the PT_MNIST_64 dataset, the buffer awaits the culmination of eight cycles to accumulate the entirety of input fields. In each batch, eight parallel multiplication and addition operations are performed for both real and imaginary components of the data. At the conclusion of the eighth cycle, the weighted sums are consolidated to the last register (for example, in Figure 5A Reg7) and relayed through the activation function—CReLU for CVNN and ReLU for RVNN. Following this, the processed data traverse a multiplier corresponding to the output layer where they are multiplied with the layer’s weights. They then undergo another round of accumulation and, subsequently, another activation function. In this process of the output layer, 20 output layer neurons are employed. These neurons, equipped with multipliers, adders, and activation functions, operate in parallel to process the data. The data flow intricacies for the FPGA implementation of the CVNNs are elucidated in Figure 5A.

The full source code is accessible via the link provided in Appendix D.

## 7. Result and Evaluation of FPGA Implementation

In this study, for the FPGA implementation of our neural network models, we employed the Virtex-7 VC707 Evaluation Platform. This platform features the xc7vx485tffg1761-2 FPGA chip, a creation of AMD Xilinx. The detailed specifications of the chip are presented as follows in Table 5.

For evaluation and reporting purposes, we did not utilize the actual FPGA hardware. Instead, we relied on the behavioral simulation and post-implementation reports provided by Xilinx Vivado.

Figure 7 presents the behavioral simulation report for CVNN_64, which refers to the CVNNs model implemented on FPGA catering to 64 data points of the polar-transformed MNIST dataset. Within the simulation window, the object termed Predicted_class displays the classification outcome for the CVNNs model tailored for the FPGA, targeting the polar-transformed MNIST dataset with 64 data points. Comparable classification outcomes emerged for the other model variations.

### 7.1. Maximum Operating Frequency ($F_{max}$)

The maximum operating frequency, denoted as $F_{\max}$, is a crucial metric derived from the Worst Negative Slack (WNS) present in Vivado’s “Timing Summary Report” post-synthesis and implementation.

In digital design, slack quantifies the deviation between the expected and actual arrival times of signals, as defined by the design’s timing constraints. A negative slack is indicative of a timing violation, suggesting that signals are not reaching their intended destinations within the desired time frame.

Of all the timing violations, the WNS represents the most pronounced delay across the entire design. A positive WNS implies that the design adheres to all its timing constraints. Conversely, a negative WNS is indicative of a breach in timing specifications, necessitating design modifications.

The effective clock period, adjusted based on the WNS, is computed as:AdjustedClockPeriod=T−WNS
where *T* is the intended clock period.

Consequently, $F_{\max}$ is derived using:$$F_{\max}=\frac{1}{\mathrm{Adjusted}\;\mathrm{Clock}\;\mathrm{Period}}=\frac{1}{T-\mathrm{WNS}}$$

For illustrative purposes, if the reported WNS is −0.5 ns and the target clock period *T* is 5 ns, the effective clock period adjusts to 5.5 ns. This results in a $F_{\max}$ of approximately 181.82 MHz, computed as $\frac{1}{5.5\;\mathrm{ns}}$.

It is worth noting that alternative approaches exist to attain the desired target frequency beyond simply recalculating $F_{\max}$ based on WNS. However, in the scope of this research, our emphasis was on determining $F_{\max}$ using the WNS. Exploring these alternative strategies might be a point of interest for future work to refine and optimize the design.

During the course of our research, while the above formula provided a theoretical maximum frequency, we opted for a slightly reduced frequency to instate a safety margin. This precaution ensures the design remains resilient against potential timing constraint violations.

Following the generation of the post-implementation timing reports for all models in Vivado, we tabulated the results, as shown in Table 6. The table summarizes the maximum clock period, the derived maximum operating frequency, and the Worst Negative Slack (WNS) for each of the models.

### 7.2. Benchmarking FPGA Inference Models against CPU and GPU Platforms

To gauge the performance of our FPGA-based inference models, we benchmarked them against CPU- and GPU-based models. For this comparison, we utilized the Google Colab platform, executing inference models implemented in the Python programming language. Specifications of the computational environment on Google Colab are given in the Table 2.

The ensuing table, Table 7, delineates the inference times in milliseconds for different models on various hardware platforms. Each model inferred a total of 10,000 MNIST test datasets on both CPU and GPU environments in Google Colab. Five trials were conducted for each model and hardware platform, and the average results are presented in the table.

#### 7.2.1. Inference Time Comparison

Table 6 provides insights into the maximum operating frequency for each implementation. To comprehensively understand the performance of our FPGA-based models, we derive the total inference time required to classify the 10,000 MNIST test dataset. The formula to compute this is delineated below:(12)Infer_time=T×N×S
where:*T*: Duration of each clock cycle in nanoseconds.*N*: Number of clock cycles needed to classify each sample.*S*: Total number of samples, which is 10,000 in this case.

Clock cycle requirements for the different models:CVNN_64 and RVNN_64 each require 8 clock cycles to classify a sample.CVNN_128 and RVNN_128 each necessitate 16 clock cycles to complete the classification of a sample.The RVNN_Raw_MNIST model, designed to process 16 input fields per clock cycle, efficiently handles the raw MNIST dataset, which comprises 784 input fields per sample. Consequently, it takes the model 49 clock cycles to classify each sample.

Following our analysis, the inferred times for each model are summarized in Table 8.

#### 7.2.2. Inference Time Comparison across CPU, GPU, and FPGA

To convey the concept of “how fast” a model is, we use the reciprocal of the inference time. In other words, we computed the “speed” as:(13)$$\mathrm{Speed}=\frac{1}{\mathrm{Inference}\;\mathrm{Time}}$$

It defines the rate at which the system processes a sample, or the amount of process completed per millisecond in this case. This gives us a measure where larger values indicate faster performance. Note that for this metric, a higher value is better, which is the opposite of the inference time where a lower value is better. As shown in Equation (Equation 13), the speed is the inverse of the inference time.

In our comparative analysis of neural network model speeds across CPU, GPU, and FPGA platforms, several distinct patterns emerged as illustrated in Figure 8. Most prominently, the FPGA demonstrated higher speeds when compared to both the CPU and GPU, underscoring its viability for tasks requiring swift computation. The GPU, with its parallel processing capabilities, showcased higher speeds relative to the CPU, signifying its prowess in neural network computations. However, it was the notable speed of the FPGA, especially for specific neural network architectures, that was most noticeable.

Diving deeper into the nuances of the model variants, the CVNN_64 and RVNN_64 models displayed considerably greater speeds compared to their CVNN_128 and RVNN_128 counterparts. This observation is intuitive: models with a reduced complexity and fewer parameters naturally lead to faster computation times. While we did not implement the RVNN_Raw_MNIST model on FPGA, when juxtaposed with the results from the FPGA models, it becomes evident that FPGAs substantially outperform the models using the original MNIST dataset.

This further suggests that the throughput for our CVNNs and RVNNs implemented on FPGA systems is enhanced compared to that of similar networks running on CPU- or GPU-based systems.

In summation, our analysis emphasizes the critical role of hardware selection in maximizing neural network efficiency. Within this context, FPGAs emerge as an optimal choice, particularly when compared to models trained on traditional datasets.

#### 7.2.3. Power Consumption Comparison

Understanding the power consumption of neural network models is crucial for their deployment in real-world scenarios, especially in power-sensitive applications. Our neural network designs, when implemented on FPGA, yielded power consumption results as detailed in Table 9.

**Interpretation of Power Metrics:** Thermal power, often termed as the total on-chip power, is a summation of dynamic and static power. Dynamic power refers to the average power consumption during logic utilization and switching activities. Conversely, static power characterizes the scenario where the device remains active but abstains from any form of utilization or switching.

From Table 9, it is evident that RVNN models are more power-efficient compared to CVNN models. Moreover, models based on 64 input entries exhibit lower power consumption than their 128 input counterparts. However, FPGA implementation of the neural network model (RVNN_Raw_MNIST) for unprocessed raw MNIST data consumed more than double the power of the most expensive neural network model (such as CVNN_128) for preprocessed data in the classification of 10,000 frames of the MNIST test dataset.

**Comparison with CPU and GPU:** Our attempt to directly compare FPGA-based power consumption with CPU and GPU systems encountered a challenge. Google Colab, which was employed for CPU- and GPU-based designs, does not provide direct power consumption metrics. However, resorting to the respective processor datasheets, we discerned their TDP (Thermal Design Power) ratings. The Intel(R) Xeon(R) CPU @ 2.00GHz employed by Colab boasts a TDP of 270 watts [28], whereas its GPU, the NVIDIA® Tesla® P4, has a TDP of 75 W [29].

Further exploration led us to test on a multi-core CPU-based laptop powered by the 11th Gen Intel(R) Core(TM) i5-1135G7 @ 2.40GHz processor. This processor’s datasheet indicates a power consumption of approximately 25 watts in its Low Power Mode [30].

Han J. et al. [21] developed a spiking neural network model for FPGA platforms, and similarly implemented it in Python for NVIDIA Tesla P100 GPU. In their experiments processing 10,000 frames from the MNIST test dataset, the GPU implementation required 7.96 s and averaged 29.6 W of power consumption. Contrastingly, our CVNN_128 model showcases better efficiency, consuming merely 2.917 watts per second and completing the same task in a swift 1.6 ms. This performance not only makes our system almost 100 times faster than the GPU-based solution but also achieves around ten times greater power efficiency.

The literature provides further insights. Research shows that contemporary laptops generally consume power in the range of 8 to 30 W [31]. Another study, utilizing a shunt resistor with a laptop’s power supply, revealed that the Intel i7-4820K processor expends between 10–80 W, contingent on the task [32].

Comparative analyses between CPU, GPU, and FPGA platforms for identical tasks have been conducted. One such study suggests that the Intel Core2 QX9650 CPU, NVidia GTX 280 GPU, and Xilinx xc5vlx330 FPGA consume maximum powers of 170 watts, 178 watts, and 30 watts, respectively [33]. Another comparison focused on energy efficiency for various vision kernels. In this study, the utilized CPU and GPU came equipped with on-board power measuring ICs. The results unequivocally demonstrated that the FPGA accelerator outperforms both GPU and CPU systems across all test cases [34].

Note: Our experiments were conducted using Google Colab, which, while supportive of basic profiling tools for TensorFlow and PyTorch, offers limited capabilities for advanced, hardware-specific profiling. Tools like NVIDIA’s nvprof, crucial for detailed GPU performance analysis, are not fully supported in Colab’s remote server environment. This posed a limitation in obtaining precise GPU power consumption measurements for our study.

#### 7.2.4. Resource Utilization among Different Neural Network Models Implemented on FPGA

This section delves into the resource utilization of various neural network models when implemented on the Virtex-7 VC707 Evaluation Platform, which features the xc7vx485tffg1761-2 FPGA chip. The neural network models under consideration include RVNN_64, CVNN_64, RVNN_128, and CVNN_128. In the context of FPGA implementations, the resources can be described as follows in Table 10:**LUT (Look-Up Table)**: Used for implementing combinational logic functions.**FF (Flip-Flop)**: Represents sequential logic, storing binary values.**DSP (Digital Signal Processor)**: Useful for performing arithmetic operations, especially multiplication.**IO (Input–Output Port)**: Interfaces for the FPGA to communicate with external components.**BUFG (Global Buffer)**: Provides clock and reset signal distribution across the FPGA.

In the analysis of the resource utilization, CVNN_128 consistently demands the most resources, particularly in LUTs and FFs, while RVNN_64 remains the least resource-intensive. DSP utilization is highest for CVNN_64 and CVNN_128, and IO consumption is fairly consistent across models, with a minor edge for CVNN_64. Notably, all models have minimal BUFG consumption, utilizing only a fraction of what is available.

#### 7.2.5. Evaluation with Existing Result

Until now, we have come across just one FPGA-based CVNNs model, published in May 2023. However, it should be noted that this model was not applied to MNIST or image classification tasks, making it incomparable to our research.

To assess our findings in the context of existing work, we selected two distinct neural network models (SNN [18] and CNN [35]) implemented on FPGA for MNIST classification. In Table 11, we present a performance comparison among these three FPGA-based neural network models for MNIST digit classification.

In recent publications, a spiking neural network (SNN) by Zhang, J. et el. [18] published in May 2023 and a convolutional neural network (CNN) by Parra, D. et el. [35] published in October 2023 achieved accuracy rates of 90.39% and 94.43%, respectively, on the MNIST test dataset. Our model, CVNN_128, achieved an accuracy of 88.3% on the same dataset. Comparing power consumption, the SNN, CNN, and CVNN_128 models consumed 1.131 W, 0.5715 W, and 0.004656 W, respectively, for the 10,000 frames of the MNIST test dataset. Despite its slightly lower accuracy, CVNN_128 stands out for its significantly lower power consumption (almost 122 times less than CNN and 240 times less than SNN) and exceptional speed (thousands of times faster than SNN and over hundred times faster than CNN).

Table 12 provides a comparison of resource utilization on FPGA for the SNN, CNN, and CVNN_128 models. It is evident that the SNN consumes more resources than the CNN and CVNN. The CNN exhibits the most efficient resource utilization among them. It is important to note that these are three distinct models, each optimized differently, making a direct comparison challenging. Nevertheless, the resource utilization table offers a general idea of the differences in their designs.

In summary, our CVNN_128 model showcases notable improvements in both energy efficiency and processing speed when compared to recently published models. We believe that the integration of polar representation for 2D images and CVNNs on FPGA holds great promise for energy-constrained environments, offering faster processing capabilities.

## 8. Discussion and Conclusions

This research set out to explore the implementation of CVNNs for polar represention of 2D image classification on FPGA. The primary objectives were to assess the effectiveness of CVNNs in this context and to evaluate the performance benefits of FPGA-based implementations.

The results indicate a notable performance in the classification accuracy of the polar-represented MNIST dataset using CVNNs. In our comparative analysis against real-valued neural networks (RVNNs), we observed that the CVNN model with 128 input data points (CVNN_128) achieved a classification accuracy 0.8 percent higher than its RVNN counterpart, RVNN_128. Furthermore, with a more reduced number of input data points (64 data points), CVNN_64 exhibited a 1.1 percent higher classification accuracy when compared to RVNN_64 in the context of processing the polar-represented MNIST handwritten digit test dataset. These findings align with our initial hypothesis, demonstrating that complex-valued networks excel in handling polar-represented image data. This superiority arises from CVNN’s ability to learn correlations between magnitude and phase information in complex data, resulting in improved performance compared to RVNNs. The use of FPGAs for implementation further enhanced the computational efficiency, showcasing the potential of hardware acceleration in neural network processing.

Comparatively, the FPGA implementation demonstrated improvements in processing speed and power efficiency. Our research reveals that our robust design, CVNN_128, is approximately 200 times faster than a CPU-based computer running CVNN and 150 times faster than a GPU-based computer for MNIST digit classification. Additionally, it demonstrates lower power consumption when compared to CPU- and GPU-based systems, as well as other neural network models implemented on FPGA for MNIST digit classification. This supports the hypothesis that FPGA-based systems can provide significant advantages in specific neural network applications, particularly in scenarios where low power consumption and high-speed computation are crucial.

### 8.1. Limitations of the Study

The study was limited to the MNIST handwritten digits dataset, which may constrain the generalizability of the findings to other types of datasets.

### 8.2. Recommendations for Future Research

Further research should extend the validation of polar-represented image classification using CVNNs beyond the MNIST dataset to include a wider array of 2D images. This expansion would test the model’s generalizability and effectiveness across diverse image sets.

Exploring the use of spiking neural networks (SNNs) for polar-represented image data also presents a valuable opportunity. A comparative analysis between SNN and CVNN performance could offer deeper insights into the potential benefits of each neural network type for specific image classification tasks since the reprocessing technique was primarily thought of for SNNs.

Finally, the optimization of CVNN implementation on FPGAs warrants a continued research effort, particularly through enhanced pipelining techniques, which could significantly improve computational throughput and energy efficiency. Such advancements could bring FPGA-based CVNNs to the forefront of practical applications, where resource optimization is paramount.

### 8.3. Conclusions

This research highlights the possibilities of using polar representation of 2D images and CVNNs through FPGA-based implementations for image classification tasks. The results provide valuable insights into the realm of neural network acceleration and pave the way for further exploration into hardware-accelerated machine learning.

## Figures and Tables

**Figure 1 sensors-24-00897-f001:**
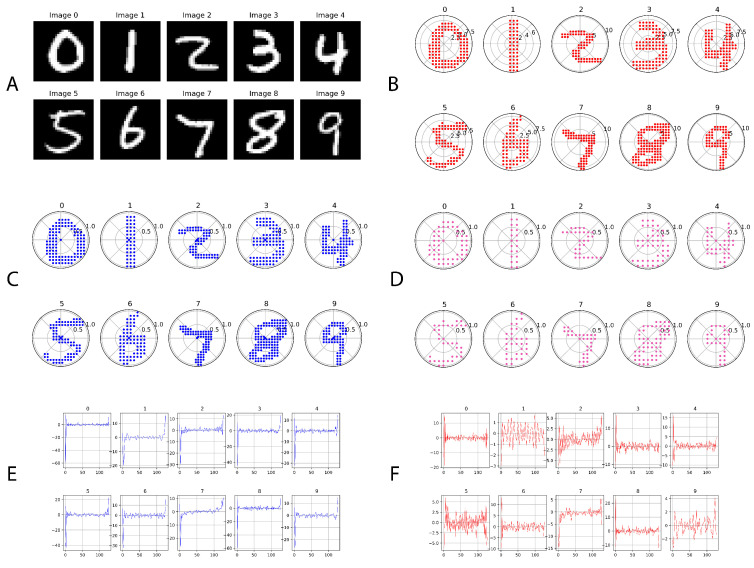
MNIST digits and their polar-transformed and serialized view. (**A**) 0–9 original MNIST handwritten digits images, (**B**) illustration of polar-transformed MNIST 0–9 images, (**C**) polar-transformed 0–9 MNIST image where N = 128, ρ = [0,1], (**D**) polar-transformed 0–9 MNIST image where N = 64, ρ = [0,1], (**E**) DFT coefficient (magnitude)—N = 128, and (**F**) DFT coefficient (phase)—N = 128.

**Figure 2 sensors-24-00897-f002:**
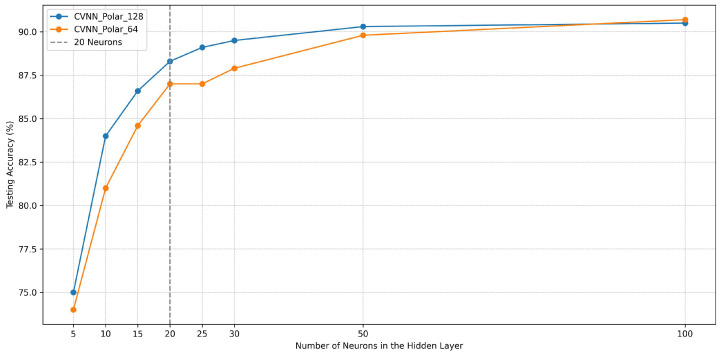
Testing accuracy vs. number of hidden layer neurons.

**Figure 3 sensors-24-00897-f003:**
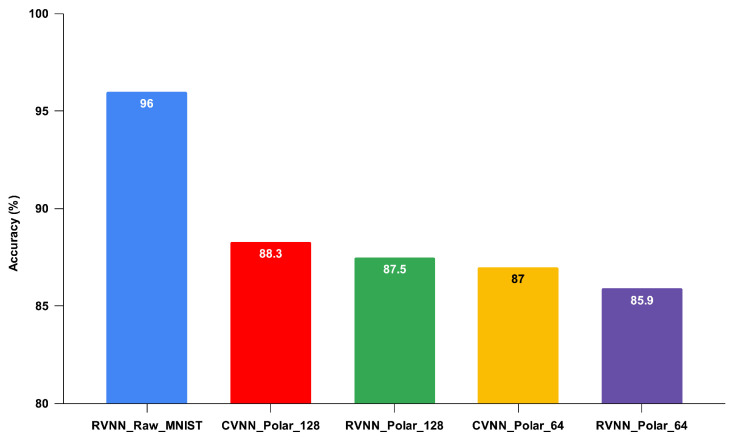
Testing accuracy of various neural network models with 20 hidden layer neurons for the MNIST dataset.

**Figure 4 sensors-24-00897-f004:**
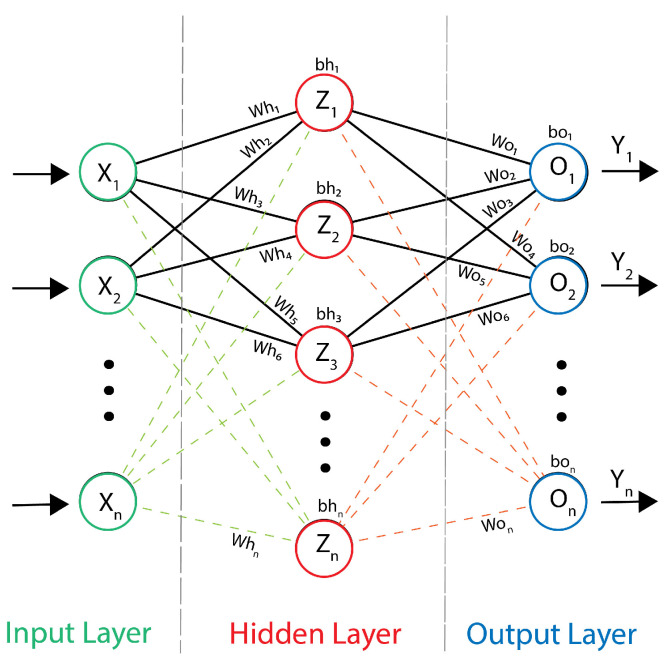
Typical inference structure of the CVNNs.

**Figure 5 sensors-24-00897-f005:**
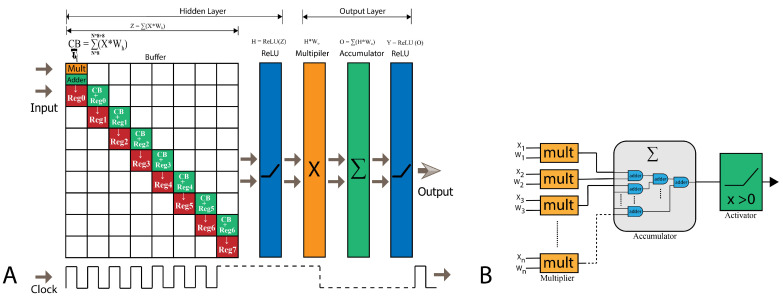
(**A**) Data flow diagram of the proposed neural network models on FPGA and (**B**) primary schematic diagram of a single neuron implemented on FPGA.

**Figure 6 sensors-24-00897-f006:**
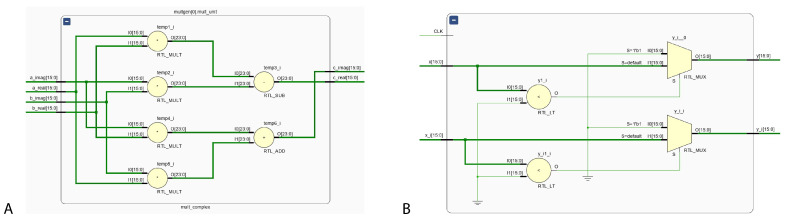
Schematic diagram of (**A**) complex-valued multiplier and (**B**) CReLU activation in FPGA.

**Figure 7 sensors-24-00897-f007:**
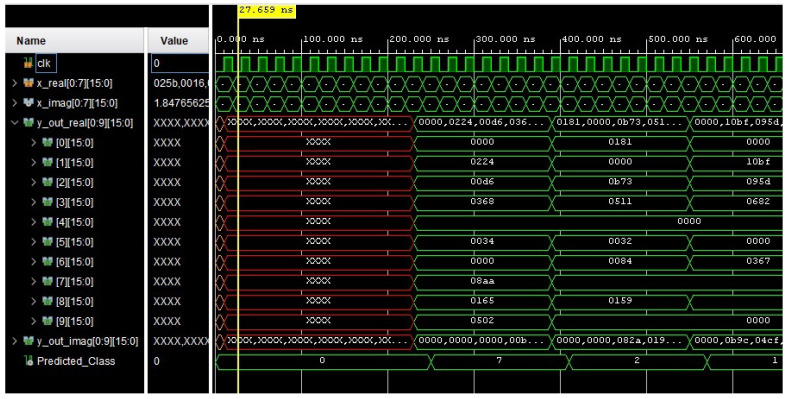
Behavioral simulation result of CVNN_64.

**Figure 8 sensors-24-00897-f008:**
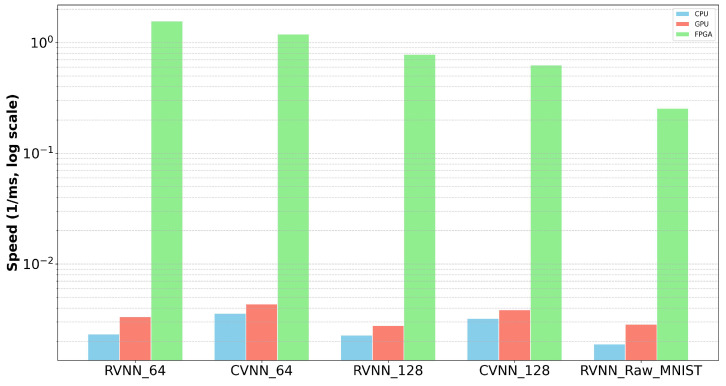
Speed across different hardware platforms (log scale).

**Table 1 sensors-24-00897-t001:** The programming language and libraries used.

Item	Description
Language	Python
Main libraries	TensorFlow and Keras
Custom library	cvnn, which provides specialized layers and functions for CVNNs.

**Table 2 sensors-24-00897-t002:** Hardware and software environment details.

Software Environment	Specification
Platform	Google Colab
Operating system	Ubuntu 22.04.2 LTS
Python version	3.10.12
TensorFlow version	TensorFlow 2.13.0
**Hardware Environment**	
System RAM	51.0 GB
CPU	Intel® Xeon® CPU @ 2.00 GHz
GPU	NVIDIA® Tesla® P4

**Table 3 sensors-24-00897-t003:** Model configurations.

Model 1: 128 Data Points	Model 2: 64 Data Points
Input layer neurons: 128	Input layer neurons: 64
Hidden layer neurons: 10, 15, 20, 25, 30, 50, 100	Hidden layer neurons: 10, 15, 20, 25, 30, 50, 100
Output layer neurons: 10	Output layer neurons: 10
Batch size = 32	Batch size = 32
Number of epochs: 50	Number of epocs: 50

**Table 4 sensors-24-00897-t004:** Dataset details.

Dataset Details	
Training samples	60,000
Testing samples	10,000
Validation split	0.02% of the training data were set aside for validation

**Table 5 sensors-24-00897-t005:** Configuration of the xc7vx485tffg1761-2 FPGA chip.

Parameter	Value
Device	xc7vx485tffg1761-2
Manufacturer	AMD Xilinx
Logic elements	485,760
DSP units	2800
I/O pins	700
Supply voltage	0.97 V–1.03 V

**Table 6 sensors-24-00897-t006:** Post-implementation timing results for different neural network models on Vivado.

Model	Max. Clock Period (ns)	Max. Freq. (MHz)	WNS (ns)
RVNN_64	8	125	0.155
CVNN_64	10.5	95.238	0.096
RVNN_128	8	125	159
CVNN_128	10	100	0.182
RVNN_Raw_MNIST	8	125	0.166

**Table 7 sensors-24-00897-t007:** Inference times in milliseconds for different models on various hardware platforms.

Models	Inference Time (CPU, ms)	Inference Time (GPU, ms)	Inference Time (FPGA, ms)	Inference Time (CPU_2, ms)
RVNN_64	430	300	0.64	180
CVNN_64	280	230	0.84	210
RVNN_128	440	360	1.28	210
CVNN_128	310	260	1.6	230
RVNN_Raw_MNIST	530	350	3.92	240

**Table 8 sensors-24-00897-t008:** Inference times for neural network models on FPGA.

Model	T × N × S	Inference Time (ns)
RVNN_64	8 × 8 × 10,000	640,000
CVNN_64	10.5 × 8× 10,000	840,000
RVNN_128	8 × 16 × 10,000	1,280,000
CVNN_128	10 × 16 × 10,000	1,600,000
RVNN_Raw_MNIST	8 × 49 × 10,000	3,920,000

**Table 9 sensors-24-00897-t009:** Power consumption for neural network models on FPGA.

Model	Thermal Power	Static Power	Power Consumption
	**(Watt per Second)**	**(Watt per Second)**	**(Watt per 10,000 Frame)**
RVNN_64	1.585	0.252	0.001014
CVNN_64	2.585	0.26	0.022167
RVNN_128	1.694	0.253	0.002168
CVNN_128	2.917	0.263	0.004656
RVNN_Raw_MNIST	2.53	0.274	0.009917

**Table 10 sensors-24-00897-t010:** Resource utilization among different neural network models.

Resource	RVNN_64	CVNN_64	RVNN_128	CVNN_128	RVNN_Raw_MNIST	Available
LUT	9123	17,723	13,122	24,164	20,993	303,600
FF	3110	5936	5703	11,520	16,066	607,200
BRAM	-	-	-	-	160	1,030
DSP	485	1333	469	1333	507	2,800
IO	417	577	417	577	417	700
BUFG	1	1	1	1	1	32

**Table 11 sensors-24-00897-t011:** Comparative analysis of different neural network models on FPGA.

Models	Accuracy	Inference Time	Power (Watt per Second)	(Watt per 10,000 Frame)
SNN	90.39%	1 s	1.131 W	1.131 W
CNN	94.43%	0.127 s	4.5 W	0.5715 W
CVNN_128	88.3%	0.0016 s	2.91 W	0.004656 W
CVNN_64	87.0%	0.00084 s	2.58 W	0.002167 W

**Table 12 sensors-24-00897-t012:** Resource utilization for different neural network models on FPGA.

Resource	SNN	CNN	CVNN_128
LUT	73,677	6373	24,164
LUTRAM	3669	71	0
FF	32,853	12,470	11,520
BRAM	0	0	0
DSP	10	93	1333
IO	419	18	577
BUFG	1	0	1

## Data Availability

Data are contained within the article.

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
