# Peer review of "FPGA Implementation of Complex-Valued Neural Network for Polar-Represented Image Classification"

_sensors, 2024, doi:10.3390/s24030897_

Round 1

Reviewer 1 Report

Comments and Suggestions for Authors

Implementing a Complex-Valued Neural Network (CVNN) for Polar-Represented Image Classification on an FPGA involves several key steps. The use of complex numbers in neural networks can be beneficial for tasks involving phase information or circular data representation, such as polar images

Few points to improve the paper.

1. The abstract should omit the GitHub reference. Instead, it ought to underscore the authors' specific contributions, particularly concerning the performance of the proposed study.

2. The paper's architecture lacks appropriate organization. Consider rearranging sections, and the '1.3 Outline' section should not stand alone; its integration should ensure a smoother flow of the paper.

3. Section 1.1, 'Related Work,' should be a distinct section. Additionally, correct the spelling to '1.1 Related Work.'

4. The current structure of the paper gives the impression of a thesis. Please overhaul the entire paper for improved coherence.

5. “4.2.1 'Objective & Hypothesis' is redundant, duplicating 1.2 'Research Objectives.'

6. In Table 2, 'Hardware and Software Environment Details,' segregate the information under distinct hardware and software headings.

7. In Table 5, 'Accuracy Comparison with contemporary research,' eliminate the surplus column.

8. Section 4.3, 'Conclusions,' raises the question of why a conclusion is necessary at this point. Section 7, 'Discussion and Conclusion,' is the primary section for conclusions, with subsection 7.3 addressing conclusions.

9. Figure 7 and Figure 8 are not visible; they need to be redrawn.

10. A comparison with previous studies and techniques is lacking, which could provide support for your work.

Comments on the Quality of English Language

Minor Edits is required 

Author Response

Response to Reviewer 1 Comments 

  1. The abstract should omit the GitHub reference. Instead, it ought to underscore the authors' specific contributions, particularly concerning the performance of the proposed study.

Authors Response To Point 1:

Thank you for your valuable suggestion regarding the abstract. We have removed the GitHub reference and revised the abstract to more clearly highlight our specific contributions and the performance outcomes of our study.

  1. The paper's architecture lacks appropriate organization. Consider rearranging sections, and the '1.3 Outline' section should not stand alone; its integration should ensure a smoother flow of the paper.

Authors Response To Point 2:

Thank you for your suggestion on improving our paper's structure. We have removed the '1.3 Outline' section and seamlessly integrated its content after the 'Related Work' section for better flow.

We appreciate your guidance in enhancing the structure and readability of our work.

  1. Section 1.1, 'Related Work,' should be a distinct section. Additionally, correct the spelling to '1.1 Related Work.'

Authors Response To Point 3:

Thank you for pointing out the needed corrections in Section 1.1. We have amended the spelling and restructured 'Related Work' as a distinct section for improved clarity and organization.

  1. The current structure of the paper gives the impression of a thesis. Please overhaul the entire paper for improved coherence.

 Authors Response To Point 4:

Thank you for your valuable feedback regarding the structure of our paper. Acknowledging your observation, we have undertaken a comprehensive restructuring of the manuscript to enhance its coherence and align it more closely with the format and style expected of a research paper. This overhaul addresses the flow, clarity, and overall presentation of the content.

  1. “4.2.1 'Objective & Hypothesis' is redundant, duplicating 1.2 'Research Objectives.'

Authors Response To Point 5:

Thank you for your observation regarding the redundancy in Section 4.2.1 'Objective & Hypothesis' and its overlap with Section 1.2 'Research Objectives'. Following your suggestion, we have removed Section 4.2.1 to streamline the content and eliminate duplication.

  1. In Table 2, 'Hardware and Software Environment Details,' segregate the information under distinct hardware and software headings.

Authors Response To Point 6:

Thank you for your recommendation regarding Table 2, 'Hardware and Software Environment Details.' We have now segregated the information into distinct hardware and software sections for clearer presentation and ease of understanding.

  1. In Table 5, 'Accuracy Comparison with contemporary research,' eliminate the surplus column.

Authors Response To Point 7:

Thank you for your feedback on Table 5, 'Accuracy Comparison with contemporary research.' Upon reevaluation, we realized that this table is indeed redundant since the same information is already comprehensively provided in the text. Therefore, we have removed the table to enhance the clarity and conciseness of the manuscript.

  1. Section 4.3, 'Conclusions,' raises the question of why a conclusion is necessary at this point. Section 7, 'Discussion and Conclusion,' is the primary section for conclusions, with subsection 7.3 addressing conclusions.

Authors Response To Point 8:

Thank you for highlighting the redundancy in the placement of conclusions within the manuscript. In line with your observation, we have removed the premature conclusions presented in Section 4.3. This ensures that the final and comprehensive conclusions are appropriately reserved for Section 8, 'Discussion and Conclusion.'

  1. Figure 7 and Figure 8 are not visible; they need to be redrawn.

Authors Response To Point 9:

Thank you for pointing out the issues with the clarity of the text in Figs. 7 and 8. We have replaced these figures with high-resolution versions to ensure better readability. However, we noticed that while the supplementary files retain their original quality, the manuscript files appear to be compressed during the upload process, affecting the image clarity.

We kindly request that the manuscript files not be compressed upon submission to preserve the quality of these images. This measure will help maintain the integrity and readability of the figures for the readers.

We appreciate your understanding and assistance in resolving this matter.

  1. A comparison with previous studies and techniques is lacking, which could provide support for your work.

Authors Response To Point 10:

Thank you for your comment on the need for a comparison with previous studies and techniques. We would like to point out that discussions on previous techniques, specifically Convolutional Neural Network (CNN) and Spiking Neural Network (SNN), are included in Sections 5.2.4 and 7.2.5 of our manuscript. However, given the fundamentally different architectures and operational paradigms of CNNs and SNNs compared to Complex-Valued Neural Networks (CVNNs), a direct structural comparison was not pursued.

Instead, to effectively benchmark and demonstrate the capabilities of our CVNN model, we implemented a Real-Valued Neural Network (RVNN) for both the polar-transformed and original MNIST data. This comparative analysis, which is detailed throughout the paper, was designed to underscore the unique strengths and applications of CVNNs in the context of the given datasets.

We believe this approach provides a more relevant and meaningful comparison, aligning with the specific focus and contributions of our work.

Reviewer 2 Report

Comments and Suggestions for Authors

The manuscript presents FPGA implementation of CVNN for polar represented image classification for the MNIST dataset consisting of 10000 test-set. The manuscript mainly claims:

-          Methodological innovation lies in Cartesian to the polar transformation of 2D

-          Optimized CVNN models suitable for FPGA

-          HW accelerator and algorithmic changes to improve the performance of

The study mentions in the paragraph “What sets our research apart” – a nuanced approach should improve the accuracy. However, the results contradict the claim with reduced accuracy (88.3% compared to 94.43%) by 6.13%. The title mentions CVNN implementation, but throughout the study, CVNN and RVNN are simultaneously discussed and implemented. The contribution of the presented work is not very clear. Multiple conclusion sections make it confusing to follow the study. One of the biggest concerns is the lack of significant HW architecture details. The HW architecture is not presented in detail and lacks optimizations to improve the performance, especially when the claim is “more than a hundred times faster”.

Major concerns:

-          The key highlight point of employing FPGA is obvious, as is the explanation, though it seems repetitive in Sec 1. Additionally, Fig. 1 also shares the same information and doesn’t necessarily convey further information. How are the resource, power, speed, and accuracy bar values obtained? Are there any references to the information depicted? If not, it may not be a good idea to show it.

-          Fig. 2, size and fonts are too small to read

-          “Our CVNN architecture is designed as a feedforward model, constructed using TensorFlow’s renowned Sequential API.” – Can you please provide a citation?

-          Repetitive textual information throughout the manuscript. For example, “CVNN_Polar_128 with 128 datapoints and CVNN_Polar_64 with 64 datapoints. Performance evaluations, visualized through accuracy trends, training curves, and loss plots, details provided in Appendix B revealed that CVNN_Polar_128 generally outperformed CVNN_Polar_64, especially with up to 20 neurons in the hidden layer.” – This point is discussed multiple times in Sec 4.

-          Why does “RVNN_Raw_MINIST” have 96% accuracy in Fig. 4 and 98% accuracy in Fig. 5?

-          Sec 5 title states FPGA implementation of CVNNs and discusses the redundant introduction and background information of FPGA and CVNN for two pages

-          Equations 2 and 20 both define CReLU – Is there a reason for defining them twice? Please check all discrepancies for the rest of the equations

-          “The complete source code for our FPGA implementations of the Complex-Valued Neural Networks (CVNNs) and Real-Valued Neural Networks (RVNNs) is provided in the Appendix.” – The appendix has no source code provided

-          The manuscript provides lots of unnecessary details, such as a long explanation of WNS and time period, among other topics. However, the focus should be on implementation details. Sec 5 uses five pages. However, roughly 1-2 pages describe HW implementation details. How are the design blocks connected? What type of memories were used? Can you elaborate on parallelism? How many MAC units were employed, and in parallel? The study significantly lacks HW architecture details

-          Table 10 shares the number of clock cycles for each model. There should be an explanation on how each model uses N number of clock cycles instead of stating direct value. How many computation process blocks are there? What is the compute clock cycle on each? What makes a total of 49 for RVNN_RAW_MNIST and the same for other models?

-          “While we did not implement the RVNN_Raw_MNIST model on FPGA, when juxtaposed with the results from the FPGA models, it becomes evident that FPGAs substantially outperform the models using the original MNIST dataset.” – How is the speed calculated/measured in Fig. 10 is not clear to readers

-          The reported power consumption for CPU and GPU present TDP values. At least for GPU, a profiler could be used to get more accurate power results (such as nvprof)

-          “Comparative analyses between CPU, GPU, and FPGA platforms for identical tasks have been conducted. One such study suggests that the Intel Core2 QX9650 CPU, NVidia GTX 280 GPU, and Xilinx xc5vlx330 FPGA consume maximum power of 170 watts, 178 watts, and 30 watts respectively [38]. Another comparison focused on energy efficiency for various vision kernels. In this study, the utilized CPU and GPU came equipped with on-board power measuring ICs. The results unequivocally demonstrated that the FPGA accelerator outperforms both GPU and CPU systems across all test cases [39].” – It is not clear why a whole paragraph on CPU, GPU, and FPGA comparison on different tasks than RVNN or CVNN is discussed in the results section 6.2.3.

Comments on the Quality of English Language

Minor concerns:

-          Section 1.1 title

-          “The approach utilizes rate coding to map ANN” – ANN is not abbreviated in the manuscript

-          OFDM is not abbreviated

-          CVNN is abbreviated in Sec 1 – Introduction, though the last paragraph in Sec 1.1 mentions “Notably, to best of our knowledge, no prior implementations of complex-valued neural networks on FPGA for MNIST dataset classification have been found in our research.” Then CVNN is once again abbreviated in Sec 1.2. Similar for FPGA. I suggest checking the consistency of abbreviations throughout the manuscript.

-          What is the reason for System RAM 51 GB? Does it imply storage?

-          Fig. 3 data is the same as presented in Fig. 5 and doesn’t share any more valuable information

-          Justification of employing FPGAs discussed in the initial Sec 1 and 2, which is repeated in Sec 5 again

-          Figs. 7,8,9 texts are blurred and too small, making it difficult for readers to follow the implementation details

-          Tables 2 and 8 could be combined

-          Tables 9 and 11 could be combined to make it easier for readers to follow

-          Try to avoid superlative words or subjective valuations, i.e., superior, remarkable, etc., being used to highlight FPGA outperforming CPU, GPU

Author Response

Response to Reviewer 2 Comments 

Major concerns:

  1. The key highlight point of employing FPGA is obvious, as is the explanation, though it seems repetitive in Sec 1. Additionally, Fig. 1 also shares the same information and doesn’t necessarily convey further information. How are the resource, power, speed, and accuracy bar values obtained? Are there any references to the information depicted? If not, it may not be a good idea to show it.

Authors Response To Point 1:

Thank you for your valuable feedback regarding the content in Section 1 and the concerns about Fig. 1. We agree with your observation that the information presented was somewhat repetitive and that Fig. 1 did not add substantial value to the discussion.

In response, we have removed Fig. 1 from the manuscript. Additionally, we have revised Section 1 to enhance its coherence and eliminate redundancy, ensuring that the key points about employing FPGA are presented clearly and concisely.

We appreciate your guidance in refining our manuscript and ensuring the clarity of our presentation. 

  1. Fig. 2, size and fonts are too small to read

Authors Response To Point 2:

Thank you for your feedback regarding Fig. 2. We understand the concern about the font size. The figure's complexity necessitated a smaller font to fit all details into a single image. We reuploaded the high resolution images and assure you that the figure's high resolution allows for clear readability upon zooming in with a PDF reader. We appreciate your understanding and are open to making necessary adjustments based on further suggestions.

  1. “Our CVNN architecture is designed as a feedforward model, constructed using TensorFlow’s renowned Sequential API.” – Can you please provide a citation?

Authors Response To Point 3:

Thank you for pointing out the need for a citation. The TensorFlow Sequential API is a well-documented feature of TensorFlow's Keras API. I have referenced the official TensorFlow documentation for this citation (TensorFlow, n.d.).

TensorFlow. (n.d.). tf.keras.Sequential. Retrieved [12 Jan 2024], from https://www.tensorflow.org/api_docs/python/tf/keras/Sequential

  1. Repetitive textual information throughout the manuscript. For example, “CVNN_Polar_128 with 128 datapoints and CVNN_Polar_64 with 64 datapoints. Performance evaluations, visualized through accuracy trends, training curves, and loss plots, details provided in Appendix B revealed that CVNN_Polar_128 generally outperformed CVNN_Polar_64, especially with up to 20 neurons in the hidden layer.” – This point is discussed multiple times in Sec 4.

Authors Response To Point 4:

Thank you for highlighting the concern regarding the repetitive textual information, particularly in Section 4. Upon re-evaluating the manuscript, I have carefully revised and streamlined the content to eliminate redundancy. The repetitive descriptions of the CVNN_Polar_128 and CVNN_Polar_64 models, as well as their performance evaluations, have been consolidated for clarity and conciseness. These revisions ensure that each section of the manuscript now presents unique and directly relevant information, enhancing the overall readability and coherence of the paper. 

  1. Why does “RVNN_Raw_MINIST” have 96% accuracy in Fig. 4 and 98% accuracy in Fig. 5?

Authors Response To Point 5:

The observed discrepancy in the accuracy of the RVNN_Raw_MINIST model between Figures 4 and 5 is attributed to the variation in the number of neurons in the hidden layer. Specifically, the model achieved a 96% accuracy rate when configured with 20 neurons in the hidden layer, as presented in Figure 4. Conversely, the accuracy improved to 98% when the model was configured with 100 neurons in the hidden layer, which is the scenario depicted in Figure 5. Furthermore, we acknowledged that the information in Figure 5 was somewhat redundant in the context of the section it was originally placed in. To streamline the content and avoid repetition, we have moved Figure 5 to Appendix C.

  1. Sec 5 title states FPGA implementation of CVNNs and discusses the redundant introduction and background information of FPGA and CVNN for two pages.

  1. Equations 2 and 20 both define CReLU – Is there a reason for defining them twice? Please check all discrepancies for the rest of the equations

Authors Response To Point 6 and 7:

Thank you for your insightful observations regarding Section 5 and the duplication of CReLU and other definitions in Equations 2 and 20.

Regarding Section 5, we acknowledge that certain introductory and background information about FPGAs and CVNNs were indeed repeated. Based on your suggestion, we have carefully revised this section, removing redundant details about the CVNN model from the CVNN implementation section. We believe that retaining the essential background information in the FPGA implementation of CVNN section is beneficial for the coherence and completeness of our manuscript.

Additionally, we have addressed the redundancy in the equations by removing the repeated definitions from the CVNN implementation section. We have also reviewed the manuscript for any similar discrepancies in other equations to ensure clarity and precision in our technical presentation.

We appreciate your guidance in enhancing the quality of our paper.

  1. “The complete source code for our FPGA implementations of the Complex-Valued Neural Networks (CVNNs) and Real-Valued Neural Networks (RVNNs) is provided in the Appendix.” – The appendix has no source code provided.

Authors Response To Point 8:

Thank you for pointing out the oversight regarding the absence of the source code in the appendix. Upon reviewing your comment, we realized that the source code for our FPGA implementations of the Complex-Valued Neural Networks (CVNNs) and Real-Valued Neural Networks (RVNNs) was indeed not included in the initial submission. We have rectified this mistake by providing a GitHub link to the complete source code in Appendix D. We appreciate your attention to detail and apologize for any inconvenience caused by this oversight.

  1. The manuscript provides lots of unnecessary details, such as a long explanation of WNS and time period, among other topics. However, the focus should be on implementation details. Sec 5 uses five pages. However, roughly 1-2 pages describe HW implementation details. How are the design blocks connected? What type of memories were used? Can you elaborate on parallelism? How many MAC units were employed, and in parallel? The study significantly lacks HW architecture details.

Authors Response To Point 9:

Thank you for your constructive feedback regarding the level of detail in our manuscript, especially in Section 5. Based on your recommendations, we have revised Section to more explicitly focus on the hardware implementation details of our CVNN_64 and CVNN_128 models.

In our research, we utilized the PT_MNIST_64 and PT_MNIST_128 dataset, where each image sample comprises 64 and 128 input fields respectively. Our models, CVNN_64 and CVNN_128, are designed to process these input fields in batches, handling eight fields for each batch. This approach is central to the operation of our hidden layers in these models.

The weighted sum in the hidden layer is computed using the formula:

Z1 = X1 × Wh1 + X2 × Wh2 + . . . + Xn × Whn + bh1

Here, each input field (X) is multiplied by its corresponding weight (W), and a bias (bh1) is added. In our implementation, for each batch, eight input fields are multiplied with eight respective weights. The resulting products are then summed up. This weighted sum is temporarily stored in a register and aggregated with subsequent batches.

CVNN_64 Model: Requires 8 clock cycles to process each batch, handling both the real and imaginary parts simultaneously.

CVNN_128 Model: Similarly, but it requires 16 clock cycles due to the larger number of input fields.

In each batch, eight parallel multiplication and addition operations are performed for both real and imaginary components of the data.

Registers are constructed using FF and LUTs. Notably, no dedicated memory blocks are used for data storage during processing in these registers.

After processing in the hidden layer, the data advances to the output layer, where we have implemented 20 neurons. These neurons, equipped with multipliers, adders, and activation functions, operate in parallel to process the data.

The entire process, including the specific operations in each batch and the overall data flow through the model, is illustrated in Figure 7 and thoroughly discussed in Section 5.5 of our research.

We appreciate the opportunity to improve our work and hope that these amendments meet the expectations of your review.

  1. Table 10 shares the number of clock cycles for each model. There should be an explanation on how each model uses N number of clock cycles instead of stating direct value. How many computation process blocks are there? What is the compute clock cycle on each? What makes a total of 49 for RVNN_RAW_MNIST and the same for other models?

Authors Response To Point 10:

Thank you for your insightful feedback. We have taken your comments into consideration and provided a detailed explanation on the computation process and clock cycle usage for each model in Sections 6.5 and 7.2.1. 

"Considering the limited number of I/O pins on our target FPGA development board, we've optimized the system to process 8 input fields per clock cycle to address this limitation. Consequently, for the PT_MNIST_64 dataset, which contains 64 input fields per sample, the system requires 8 clock cycles, and for the PT_MNIST_128 dataset with 128 input fields per sample, it necessitates 16 clock cycles to fully process all input fields."

"The RVNN_Raw_MNIST model, designed to process 16 input fields per clock cycle, efficiently handles the raw MNIST dataset, which comprises 784 input fields per sample. Consequently, it takes the model 49 clock cycles to classify each sample."

  1. “While we did not implement the RVNN_Raw_MNIST model on FPGA, when juxtaposed with the results from the FPGA models, it becomes evident that FPGAs substantially outperform the models using the original MNIST dataset.” – How is the speed calculated/measured in Fig. 10 is not clear to readers

Authors Response To Point 11:

Thank you for your observation regarding the implementation of the RVNN_Raw_MNIST model on FPGA and the clarity of speed measurement in Figure 10. Upon review, we realized that the statement about the non-implementation of the RVNN_Raw_MNIST model on FPGA was erroneously included in our manuscript. We have corrected this error by removing the sentence from the document.

Additionally, to clarify the measurement of speed as depicted in Figure 10 (Figure 8 now), we define it as the rate at which the system processes a sample, or the amount of process completed per millisecond. In essence, this is inversely related to the inference time, which refers to the duration required to process a single sample. We have updated this information to the manuscript. We hope this elaboration resolves any ambiguities and provides a clearer understanding of our methods and results.

  1. The reported power consumption for CPU and GPU present TDP values. At least for GPU, a profiler could be used to get more accurate power results (such as nvprof).

Authors Response To Point 12:

Thank you for your valuable suggestion regarding the use of a profiler for obtaining more accurate power consumption results, particularly for GPU. We acknowledge the limitation in our methodology in this regard. Our neural network models were executed on Google Colab, which provided us with access to GPU resources. However, due to the nature of this cloud-based platform, our ability to extract detailed GPU performance information or employ tools like nvprof for precise power measurements was constrained. This limitation has been noted, and we appreciate your understanding of the challenges we faced in accessing more granular power consumption data in our experimental setup.

  1. “Comparative analyses between CPU, GPU, and FPGA platforms for identical tasks have been conducted. One such study suggests that the Intel Core2 QX9650 CPU, NVidia GTX 280 GPU, and Xilinx xc5vlx330 FPGA consume maximum power of 170 watts, 178 watts, and 30 watts respectively [38]. Another comparison focused on energy efficiency for various vision kernels. In this study, the utilized CPU and GPU came equipped with on-board power measuring ICs. The results unequivocally demonstrated that the FPGA accelerator outperforms both GPU and CPU systems across all test cases [39].” – It is not clear why a whole paragraph on CPU, GPU, and FPGA comparison on different tasks than RVNN or CVNN is discussed in the results section 6.2.3.

Authors Response To Point 13:

Thank you for your comment regarding the inclusion of a comparative analysis between CPU, GPU, and FPGA platforms in Section 6.2.3. The purpose of this discussion is to contextualize the feasibility and efficiency of FPGA-based neural network models in comparison to those run on other platforms. Our hypothesis posits that the reconfigurable and parallel nature of FPGAs can support the implementation of neural network models more rapidly and efficiently than when these models are run on traditional computing platforms. While the performance aspects of these platforms are discussed in other sections, Section 6.2.3 aims to extend this comparison to power consumption, highlighting the overall efficiency of FPGA in neural network applications. This comparative analysis is critical to understanding the broader implications of our research in the context of power efficiency in neural network implementations.

Comments on the Quality of English Language

Minor concerns:

  1. Section 1.1 title

Corrected

  1. “The approach utilizes rate coding to map ANN” – ANN is not abbreviated in the manuscript

Corrected

  1. OFDM is not abbreviated

Corrected

  1. CVNN is abbreviated in Sec 1 – Introduction, though the last paragraph in Sec 1.1 mentions “Notably, to best of our knowledge, no prior implementations of complex-valued neural networks on FPGA for MNIST dataset classification have been found in our research.” Then CVNN is once again abbreviated in Sec 1.2. Similar for FPGA. I suggest checking the consistency of abbreviations throughout the manuscript.

Corrected

  1. What is the reason for System RAM 51 GB? Does it imply storage?

Thank you for your inquiry regarding the specified 51 GB of System RAM in our Google Colab Pro setup. I would like to clarify that this figure refers to the available system memory (RAM) provided by the Google Colab Pro service, and not to storage capacity. 

We appreciate your attention to detail and your query which allowed us to clarify this aspect of our research setup. 

  1. Fig. 3 data is the same as presented in Fig. 5 and doesn’t share any more valuable information

Thank you for your valuable comment regarding Fig. 3 and Fig. 5. Upon review, we recognize that while Fig. 5 contains slightly more information than Fig. 3, its inclusion in the main body of the paper may lead to redundancy.

To address this, we have decided to move Fig. 5 to the appendix. This relocation allows readers interested in the additional details to access them without overloading the main content with repetitive information. We believe this adjustment maintains the paper’s conciseness while preserving the comprehensive nature of our research.

We are grateful for your keen observation and helpful suggestion.

  1. Justification of employing FPGAs discussed in the initial Sec 1 and 2, which is repeated in Sec 5 again

Thank you for highlighting the repetition in the justification of employing FPGAs across Sections 1, 2, and 5. Following your observation, we have carefully reviewed these sections and removed the redundant content. This edit ensures that each section now presents unique and relevant information without overlap, contributing to the overall coherence and conciseness of the paper.

We are grateful for your attentive review and guidance, which have significantly contributed to improving the quality of our manuscript.

  1. Figs. 7,8,9 texts are blurred and too small, making it difficult for readers to follow the implementation details

Thank you for pointing out the issues with the clarity of the text in Figs. 7, 8, and 9. We have replaced these figures with high-resolution versions to ensure better readability. However, we noticed that while the supplementary files retain their original quality, the manuscript files appear to be compressed during the upload process, affecting the image clarity.

We kindly request that the manuscript files not be compressed upon submission to preserve the quality of these images. This measure will help maintain the integrity and readability of the figures for the readers.

We appreciate your understanding and assistance in resolving this matter.

  1. Tables 2 and 8 could be combined

Thank you for your suggestion regarding Tables 2 and 8. Following your advice, we have combined these tables for a more streamlined presentation in the manuscript.

  1. Tables 9 and 11 could be combined to make it easier for readers to follow

Thank you for your insightful suggestion regarding Tables 9 and 11. Upon reevaluation, we recognize that Table 9 is indeed redundant as its content is effectively covered in Table 11. To streamline the manuscript and avoid repetition, we have removed Table 9. 

  1. Try to avoid superlative words or subjective valuations, i.e., superior, remarkable, etc., being used to highlight FPGA outperforming CPU, GPU

Thank you for your feedback. We have revised the manuscript to replace superlative terms with more objective language in the comparative analysis of FPGA, CPU, and GPU.

Round 2

Reviewer 2 Report

Comments and Suggestions for Authors

Thank you for clarifying the questions.

Comments on the Quality of English Language

Thank you for your answers on minor queries.